

# The energetic cost of mounting an immune response for Pallas's long-tongued bat (*Glossophaga soricina*)

Lucia V. Cabrera-Martínez[1,*], L. Gerardo Herrera M.[2,*] and Ariovaldo P. Cruz-Neto[1,*]

[1] Departamento de Zoologia, Instituto de Biociências, Universidade Estadual Paulista Júlio de Mesquita Filho, Rio Claro, São Paulo, Brasil

[2] Estacion de Biologia Chamela, Instituto de Biología, Universidad Nacional Autónoma de México, San Patricio, Jalisco, México

[*] These authors contributed equally to this work.

## ABSTRACT

The acute phase response (APR) is the first line of defense of the vertebrate immune system against pathogens. Mounting an immune response is believed to be energetically costly but direct measures of metabolic rate during immune challenges contradict this assumption. The energetic cost of APR for birds is higher than for rodents, suggesting that this response is less expensive for mammals. However, the particularly large increase in metabolic rate after APR activation for a piscivorous bat (*Myotis vivesi*) suggests that immune response might be unusually costly for bats. Here we quantified the energetic cost and body mass change associated with APR for the nectarivorous Pallas's long-tongued bat (*Glossophaga soricina*). Activation of the APR resulted in a short-term decrease in body mass and an increase in resting metabolic rate (RMR) with a total energy cost of only 2% of the total energy expenditure estimated for *G. soricina*. This increase in RMR was far from the large increase measured for piscivorous bats; rather, it was similar to the highest values reported for birds. Overall, our results suggest that the costs of APR for bats may vary interspecifically. Measurement of the energy cost of vertebrate immune response is limited to a few species and further work is warranted to evaluate its significance for an animal's energy budget.

## INTRODUCTION

Mounting an immune response is believed to be energetically costly, requiring trade-offs with other important biological functions (*Sheldon & Verhulst, 1996*). The energy cost of immune response activation by vertebrates, for example, has been hypothesized to equal that of reproduction and growth (*Lochmiller & Deerenberg, 2000*). The acute phase response (APR), in particular, is believed to be the most energetically costly component of the activation of the immune system, and therefore more prone to trade-off with other energetically expensive life-history traits (*Lochmiller & Deerenberg, 2000*; *Bonneaud et al., 2003*; *Lee, 2006*; but see *King & Swanson, 2013*). The APR, thought to be taxonomically

Corresponding author
L. Gerardo Herrera M., gherrera@ib.unam.mx

conserved amongst vertebrates, is the first line of defense of the immune system against pathogens and involves leukocytosis, fever, increased resting metabolic rate (RMR) and decreased body mass ($M_b$) (*Cray, Zaias & Altman, 2009*). However, direct measures of metabolic rate challenge the idea that an immune response is an energetically costly process. For example, RMR of several bird species increased only 5–15% following activation of humoral and cell-mediated immunities (*Hasselquist & Nilsson, 2012*).

One method used to trigger APR in vertebrates is the use of a lipopolysaccharide (LPS; *Alexander & Rietschel, 2001*). This antigen mimics a bacterial infection without actually infecting an animal with a pathogen. LPS induces an inflammatory response by increasing the release of cytokines a few hours after inoculation causing a short-term response (*Bonneaud et al., 2003*; *Demas et al., 2011*). The metabolic cost of this short-term (within 24 h after LPS injection) APR has been measured for a handful of species with contrasting results. APR activation resulted in a large increase (185%) in RMR for fish-eating Myotis (*Myotis vivesi*; *Otálora-Ardila et al., 2016*; *Otálora-Ardila et al., 2017*), a modest increase (∼26–40%) for Pekin ducks (*Anas platyrhynchos*; *Marais, Maloney & Gray, 2011*) and house sparrows (*Passer domesticus*; *King & Swanson, 2013*; *Martin et al., 2017*), a small increase (∼10–14%) for zebra finches (*Taeniopygia guttata*; (*Burness et al., 2010*), house sparrows (*Martin et al., 2017*) and the brown rat (*Rattus norvegicus*; *MacDonald et al., 2012*), and null increase for zebra finches (*Sköld-Chiriac et al., 2014*), house sparrows (*Martin et al., 2017*) and house mice (*Mus musculus*; *Baze, Hunter & Hayes, 2011*). These studies involved both captive-raised (*Burness et al., 2010*; *Baze, Hunter & Hayes, 2011*; *Marais, Maloney & Gray, 2011*; *Sköld-Chiriac et al., 2014*) and wild populations (*King & Swanson, 2013*; *Otálora-Ardila et al., 2016*; *Otálora-Ardila et al., 2017*; *Martin et al., 2017*).

A strong immune response is assumed to be more likely for long-lived animals, such as bats (*Lochmiller & Deerenberg, 2000*). Bats are one of the most diverse orders of vertebrates both in taxonomic and ecological terms and thus represent an exceptional model to test if APR is an energetically costly event. Although bats may share several features of the immune systems with other vertebrates, they do have marked qualitative and quantitative differences in their immune system (*Baker, Schountz & Wang, 2013*). Evidence suggests that the magnitude of their immune response may vary as a function of body mass and physiological and ecological factors (*Christe, Arlettaz & Vogel, 2000*; *Allen et al., 2009*; *Schneeberger, Czirják & Voigt, 2013b*; *Schneeberger, Czirják & Voigt, 2014*; *Strobel, Becker & Encarnação, 2015*). For example, APR triggered an increase in total leukocyte numbers and a decrease in $M_b$ for the short-tailed fruit bat (*Carollia perspicillata*; *Schneeberger, Czirják & Voigt, 2013a*) and an increase in total leukocyte and neutrophill numbers and no change in $M_b$ for the wrinkle-lipped bat (*Chaerephon plicatus*; *Weise et al., 2017*). APR induced a significant decrease in $M_b$ and an increase in body temperature ($T_b$) for the fish-eating Myotis (*Otálora-Ardila et al., 2016*; *Otálora-Ardila et al., 2017*), whereas for the Pallas's mastiff bat (*Molossus molossus*) there was a reduction in $M_b$ but no change in total leukocyte numbers or $T_b$ (*Stockmaier et al., 2015*). The large increase in RMR that accompanied the immune response of the fish-eating Myotis (*Otálora-Ardila et al., 2016*; *Otálora-Ardila et al., 2017*) is unusual among vertebrates but there is no available information to evaluate if this elevated RMR is common to bats. Alternatively, and consistent with the responses

reported for other aspects of bat APR, the metabolic cost of this response might vary within the order Chiroptera.

Here, we measure RMR and $M_b$ of the nectarivorous Pallas's long-tongued bat (*Glossophaga soricina*, Pallas 1766; Phyllostomidae) before and after challenging its immune system with an injection of LPS. We aimed to quantify and describe the magnitude of the energetic cost associated with APR for this plant-eating bat. We compared the effect of the APR on RMR and $M_b$ for *G. soricina* with that for the fish-eating Myotis and other vertebrates to test the hypothesis that activation of this immune response is unusually costly for bats. We are aware that our comparison is limited by the variety of protocols used in previous measurements of APR activation in vertebrates and we are therefore cautious to consider it in a qualitative framework.

## MATERIALS & METHODS

### Animal capture and housing

Adult non-reproductive individuals of *G. soricina* (nine males and four females; $M_b = 10.3 \pm 0.3$ g, mean $\pm 1$ S.E. here and thereafter) were captured with mist nets during late spring at the entrance of the El Salitre cave, 3.6 km South of Los Ortices (19°04′N, 103°43′W), Colima, Mexico. Three to four individuals were captured during each visit to the cave, and kept in a $3 \times 3 \times 3$ m flight cage in a nearby facility with natural photoperiod and temperature. They were fed a mixture of cereal, table sugar, powdered milk and banana diluted with water. Experiments commenced 2–3 days after capture and, once measurements were complete, bats were released at the site of capture. Each bat was measured only once and held in captivity for ≤5 days. We followed the American Society of Mammologists animal guidelines (*Sikes, Gannon & The Animal Care and Use Commitee of the American Society of Mammalogists, 2011*), and all protocols were performed under a scientific's collector license (FAUT-0069) granted to LGHM by the Secretaría de Medio Ambiente y Recursos Naturales, Mexico.

### Immune challenge

We challenged the immune system of bats (seven individuals) by injecting 50 μL of a 0.56 mg ml$^{-1}$ solution of LPS (L2630, Sigma-Aldrich, USA) in phosphate buffered saline (PBS; P4417, Sigma-Aldrich, USA). This is equivalent to a dose of $2.84 \pm 0.04$ mg LPS kg$^{-1}$. LPS was injected subcutaneously and the surrounding skin was sterilized with ethanol prior to and after the injection. Pilot experiments indicated that this dose was sufficiently high to elicit a sustained and significant response in RMR; lower doses did not cause a measurable response and higher doses elicited a blunted response. A control group of six individuals was injected with only PBS.

### Metabolic measurements

The energetic cost associated with mounting an immune response was indirectly assessed by measuring the rates of oxygen consumption ($\dot{V}O_2$) for individuals in post-absorptive state prior to and after receiving LPS or PBS. Experiments started at 06:00–07:00 am, with measurements of pre-injection levels for two hours, and one individual bat per trial. Pilot

experiments indicated that bats usually settled down in the chamber after 1 h, so this period was considered sufficient for RMR to achieve a steady-state that could be used as standard for comparing the incremental responses associated with the administration of LPS or PBS (see results). After this period, we removed the bat from the respirometric chamber, injected it with either LPS or PBS, and returned it to the chamber, after which $\dot{V}O_2$ was continuously measured for 8–10 h. The time required to remove the bat from the chamber, inject it and return it the chamber was less than five minutes.

We used open-flow respirometry to measure $\dot{V}O_2$ (*Voigt & Cruz-Neto, 2009*). Bats were weighed to the nearest 0.1 g (Ohaus Precision Balance, Parsippany, NJ, USA), and placed in a 300 ml cylindrical metabolic chamber. Because the chamber walls were made of smooth material, bats could hang only from plastic mesh placed at the top. A similar-sized, but empty, chamber was used for baseline measurements. Incurrent air, dried with a column of Drierite$^{TM}$ (W. A. Hammond Drierite, Xenia, OH, USA) was pushed through both chambers by two sets of aquarium pumps at a rate of 270–300 ml min$^{-1}$. Experiments started by taking a 10-minute baseline reading from the empty chamber and then 2 continuous 60-minute readings of the excurrent air from the experimental chamber, interspaced by a 10-minute baseline reading between each run. Excurrent air was passed through a column of Drierite$^{TM}$ then through an $O_2$ analyzer (Sable System Field Metabolic System, Sable Systems International, Las Vegas, NV, USA). Bats were removed from the chamber, injected with either PBS or LPS, and returned to the chamber. We then recommenced recording $\dot{V}O_2$ with continuous readings from the experimental chamber, interspaced by 10-minutes reading of the empty chamber at every hour.

Excurrent $O_2$ values were recorded every second, and these data subsequently analyzed by the software Expedata 1.7.2 (Sable Systems International). Readings from the $O_2$ channels were smoothed before the analysis, and $VO_2$ were calculated using equation 10.2 from *Lighton (2008)*. We calibrated the $O_2$ sensors every second day with a 20.95% $O_2$ gas mix (Praxair, Danbury, CT, USA).

The temperature ($T_a$) within the chambers was measured every 30 min with a temperature logger placed in the bottom of the chamber (I-button, Maxim Corp, San Jose, USA). $T_a$ was 28.9 ± 0.1 °C, with a minimum-maximum variation of 2.1 ± 0.1 °C, during the experiments. The mean value was slightly below the lower critical limit of the thermoneutral zone described for our focal species (31.4 °C; *Cruz-Neto & Abe, 1997*), and the maximum $T_a$ attained during a given experiment (32.6 °C) was below its upper critical temperature (35.2 °C; *Cruz-Neto & Abe, 1997*).

### Data handling and analysis

Due to the small sample size for females (two in each treatment group), data from both sexes were pooled. $M_b$ was measured within 10 min of injection of LPS or PBS ($M_{bi}$) and again after the last $\dot{V}O_2$ measurement ($M_{bf}$). Differences in mean $M_b$, as well as in absolute ($M_{bf} - M_{bi}$) and relative [($M_{bf} - M_{bi} / M_{bi}$)] changes between groups were analyzed by $t$-tests.

Activity of the bats inside the chamber caused $O_2$ levels to fluctuate during experiments. To minimize such fluctuations, we used the nadir function (Expedata 1.7.2; Sable Systems

International), which calculates the lowest-magnitude section of a specified number of contiguous data to select the lowest and most constant 15 min period of $\dot{V}O_2$ for each hour of the experiment, and then an average of these 15 min periods was used to characterize the $\dot{V}O_2$ for each hour time bin. These values were transformed to metabolic rate (MR) in kJ h$^{-1}$, by using the formula provided by *Lighton (2008)*: MR = $\dot{V}O_2 \times [16 + 5.164\,(RQ)]$. Since we did not measure $\dot{V}CO_2$, we assumed an RQ of 0.80, and used the RMR values in all subsequent analyses. We used this RQ because it is the value measured for *G. soricina* in post-absorptive state (*Welch Jr, Herrera & Suarez, 2008*).

We used a generalized linear mixed effect model to test if $T_a$ varied with time and treatment. Although different individuals were used in the PBS and LPS treatments, we had repeated measurements of $T_a$ over time for each treatment. Thus, we included bat ID as a random factor in these analyses. We tested the effects of time and treatment on RMR with similar generalized linear mixed effect models before and after the injection. For the RMR data obtained after the injection, we used net values of mass-specific RMR obtained by subtracting the lowest mass-specific RMR value before the injection from the post-injection mass-specific RMR for each individual at each time bin. Pairwise comparisons were made using a Holm-Sidak post-hoc test when the model identified significant differences.

We calculated an energetic cost index (EC) for responses to LPS and PBS to estimate the energetic cost associated with APR activation. EC is equal to the integral area under the curve that describes the variation in net mass-specific RMR after injection for each treatment. We calculated the area under the curve using the trapezoid method (*Tai, 1994*). Mass-corrected EC estimated for each treatment were compared to zero using one-sample t-tests, and the effect of $T_a$ on EC was assessed by simple linear regression for each treatment. We set the level of significance to $p \leq 0.05$ for all statistical analyses. Data analyses were performed using SigmaStat ver. 4 (Systat Sofware Inc., San Jose, CA, USA) and statistiXL ver 2.0 (statistiXL, Broadway-Nedlands, Australia).

# RESULTS

## Body mass changes

The $M_{bi}$ of *G. soricina* did not vary between treatments ($t_{11} = 0.63$, $p = 0.54$; Table 1), but $M_{bf}$ did ($t_{11} = 2.92$, $p = 0.03$). Bats injected with LPS lost more body mass than bats injected with PBS, both in absolute (PBS: $-0.6 \pm 0.11$ g, LPS: $-1.15 \pm 0.14$ g; $t_{11} = 3.05$, $p = 0.01$) and in relative (PBS: $-0.06 \pm 0.01$, LPS: $-0.11 \pm 0.01$; $t_{11} = 3.70$, $p = 0.004$) terms.

## Metabolic Rate

$T_a$ varied with time ($F_{9,129} = 4.94$; $p < 0.001$). However, it did not differ between treatments (LPS: $28.9 \pm 0.1$ °C; PBS: $29.0 \pm 0.1$ °C: $F_{1,129} = 2.75$; $p = 0.1$) or as a function of the interaction between time and treatment ($F_{9,129} = 4.94$; $p = 0.91$). Therefore, we assumed that $T_a$ measured during the experiments was not a confounding factor.

Metabolic rate measured prior to injection varied with time ($F_{1,23} = 61.9$, $p < 0.001$; Fig. 1), with the lowest value for both groups recorded 1 h before injection. This value did not differ between the two treatment groups (PBS: $0.031 \pm 0.005$ kJ h$^{-1}$ g$^{-1}$; LPS:

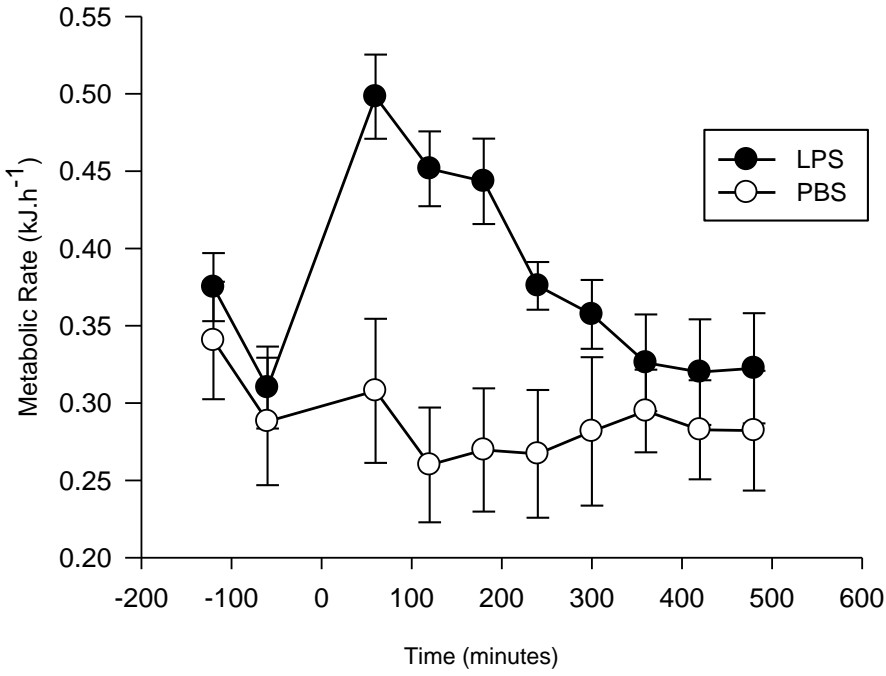

**Figure 1** **Metabolic response of the Pallas's long-tongued bat *Glossophaga soricina* after LPS and PBS administration.** Resting metabolic rate of the Pallas's long-tongued bat *Glossophaga soricina* before and after LPS and PBS administration. Injection was administered at 0 min.

**Table 1** **Initial ($M_{bi}$) and final ($M_{bf}$) body mass of bats in the two treatment groups (LPS or PBS).**

| Treatment | $M_{bi}$ | $M_{bf}$ | AD | RD |
|---|---|---|---|---|
| LPS | $10.05 \pm 0.62$ | $8.90 \pm 0.50$ | $-1.15 \pm 0.14$ | $-0.11 \pm 0.01$ |
| ($n = 7$) | (8.6–13.2) | (7.8–11.6) | ($-0.60$–$-1.62$) | ($-0.07$–$-0.15$) |
| PBS | $10.61 \pm 0.62$ | $10.01 \pm 0.66$ | $-0.60 \pm 0.11$ | $-0.06 \pm 0.01$ |
| ($n = 6$) | (8.8–12.8) | (7.9–12.0) | ($-0.30$–$-0.90$) | ($-0.03$–$-0.10$) |

**Notes.**
$M_b$ values are in grams.
AD, absolute difference ($M_{bf} - M_{bi}$) and relative difference [($M_{bf} - M_{bi}$)/ $M_{bi}$].
Values are presented as mean $\pm$ 1. s.e.m. Numbers in parenthesis denote range of observations.

$0.036 \pm 0.005$ kJ h$^{-1}$g$^{-1}$; $F_{1,23} = 0.14$, $p = 0.71$), and there was no significant treatment by time effect on pre-injection RMR ($F_{1,23} = 0.55$, $p = 0.58$). Thus, we used the RMR values obtained 1 h before the injection as our standard for calculating the net RMR after injection.

After injection, the net RMR varied significantly as a function of time ($F_{7,103} = 11.7$, $p < 0.01$; Fig. 1) and treatment ($F_{1,103} = 4.4$, $p = 0.05$; Fig. 1). The interaction term was significant ($F_{7,103} = 9.6$, $p < 0.001$), with the increase evoked by LPS being greater than the increase evoked by PBS, until 4 h after injection. After this period, no difference was observed between the net RMR of LPS and PBS injected bats ($p > 0.05$ for all pairwise comparisons).

The EC estimated for bats with the LPS treatment ($0.08 \pm 0.02$ kJ g$^{-1}$; $0.73 \pm 0.17$ kJ) was significantly different from zero ($t_6 = 3.4$, $p = 0.01$), while the EC for the PBS group during the same time period ($-0.01 \pm 0.02$ kJ g$^{-1}$; -0.13 $\pm$ 0.08 kJ) was not ($t5 = 0.5$; $p = 0.62$). Mass-specific EC was not affected by T$_a$ (LPS: $r_2 = 0.02$; $F_{1,6} = 0.11$; $p = 0.75$. PBS: $r2 = 0.02$; $F_{1,5} = 0.09$; $p = 0.78$).

## DISCUSSION

APR activation incurred an energetic cost for *Glossophaga soricina* as indicated by a short-term decrease of body mass and an increased RMR. However, the increase of RMR after LPS injection for this species was similar to the highest values measured in birds and it did not approximate that previously observed for the fish-eating Myotis.

Body mass loss ($\Delta M_b$) of *G. soricina* challenged with LPS was nearly double that observed for individuals challenged only with PBS. We found a 11% decrease of body mass in LPS challenged individuals of *G. socirina* after 8 h, similar to changes reported for piscivorous (*M. vivesi*: 8% decrease 11 h after injection; *Otálora-Ardila et al., 2016*) and insectivorous bats (*M. molossus*: 7% decrease 24 h after injection; *Stockmaier et al., 2015*), and the bird *P. domesticus* (7% decrease 16 h after injection; *Bonneaud et al., 2003*). However, it was higher than mass changes reported for other birds and rodents including *M. musculus* (no change 8 h after injection; *Baze, Hunter & Hayes, 2011*), *T. guttata* (1–3.5% decrease 24 h after injection; *Burness et al., 2010*), *Zonotrichia leucophrys* (1–5.5% decrease 24 h after injection; *Owen-Ashley et al., 2008*), *Calidris canutus* (<1% decrease 22 h after injection; *Buehler et al., 2009*) and *R. norvegicus* (4% decrease 24 h after injection; *MacDonald et al., 2012*). Although some of the difference in the magnitude of $\Delta M_b$ probably reflects differences in the dose used (see below) and in the time over which it was measured (8–24 h), it mainly reflects the mobilization of nutrient stores to cover the energetic cost associated with mounting an immune response and, thus, can be regarded as a universal component associated with the APR in vertebrates (*Ashley & Wingfield, 2012*).

Pre-injection RMR did not differ between *G. soricina* assigned to the LPS or PBS treatments, but it decreased with time as expected; manipulation of bats before placing them in the chamber usually leads to an elevated metabolic rate which decreases as bats settle down in their new environment (*Voigt & Cruz-Neto, 2009*). The average pre-injection RMR (pooled for both treatments) measured during the first (0.37 kJ h$^{-1}$) and second (0.32 kJ h$^{-1}$) hours before injection is consistent with basal metabolic rate measured for this species (0.35 kJ h$^{-1}$, range: 0.31–0.42 kJ h$^{-1}$; *Cruz-Neto & Abe, 1997*). Mean RMR 1 h after LPS injection increased by $63.2 \pm 12.8\%$ with respect to the mean value before injection compared with $14.6 \pm 15.5\%$ for bats injected with PBS. However, this is difficult to interpret as the cost of mounting an immune response as it is possible that bats entered torpor after PBS injection which may have overestimated the difference in RMR when compared with LPS-treated bats. Unfortunately, we cannot further explore this hypothesis because we did not record T$_b$ but *G. soricina* enters diurnal torpor only when food intake is restricted (*Kelm & Helversen, 2007*). Food intake was not limited the night before injections and thus we assume that bats remained normothermic during the experiments.

There are a few studies in which RMR was measured for mammals and birds within 24 h of LPS injection, and the results are quite diverse. Some studies reported no or minimal increase in RMR (*Baze, Hunter & Hayes, 2011*, *M. musculus*; *Sköld-Chiriac et al., 2014*, *T. guttata; Martin et al., 2017*, *P. domesticus*), others reported increases in RMR of 10–14% (*MacDonald et al., 2012*, *R. norvegicus;* Burness, Armstrong & Tilman-Schindel (2010), *T. guttata; Martin et al., 2017 P. domesticus*), and some reported increases of up to 26–40% (*King & Swanson, 2013*; *Martin et al., 2017*, *P. domesticus*; *Marais, Maloney & Gray, 2011*, *Anas platyrhynchos*). For *M. vivesi*, RMR increased by 185% post-injection (*Otálora-Ardila et al., 2016*). Unfortunately, results of these studies are not strictly comparable. For example, mass-specific doses vary from 0.1 to 5 mg LPS kg$^{-1}$ in these studies. Although high doses seem to elicit high responses (see also *King & Swanson, 2013*), there are discrepancies with highly variable responses both within and between species, even from the same batch and dose of LPS (*Demas et al., 2011*; *Martin et al., 2017*). In addition, measurement of RMR was conducted at different time periods after LPS injection and varying metrics used to calculate the effect on RMR compared to control or baseline data. Some studies use a fixed, and sometimes unique time period for RMR measured, and assess the immune cost by dividing a single RMR value post injection by RMR pre-injection, or by RMR measured at the same time after PBS injection.

We estimated the daily energy expenditure of *G. soricina* as 35.40 $\pm$ 1.69 kJ day$^{-1}$ using an allometric equation derived from field metabolic rate data for bats (kJ day$^{-1}$ = 5.73M$_b$(g)$^{0.79}$; *Speakman & Król, 2010*) and the M$_{bi}$ of individuals treated with LPS. The total cost associate to APR (0.73 kJ) represents 2% of the total daily energy expenditure of *G. soricina*. In contrast, *Otálora-Ardila et al. (2016)* reported that the energetic cost of APR for the fish-eating Myotis was 12%–15% of its daily energy expenditure. The overall mass-specific cost of the APR response for *G. soricina* (0.08 kJ g$^{-1}$) was less than one third of that estimated for the fish-eating Myotis (0.23 kJ g$^{-1}$).

*Voigt, Kelm & Visser (2006)* calculated that *G. commissarisi* (mean M$_b$ = 8.7 g) consumes about 0.193 kJ per visit to a flower, and to cover its daily energy expenditure (45.7 kJ day$^{-1}$) it would need either to monopolize 26–90 plants or visit roughly 236 flowers per night. A similar calculation for *G. soricina* suggests that this species would need to visit 183 flowers to meet its daily energy expenditure. The energetic cost of APR would require *G. soricina* to monopolize one additional plant or visit an additional 3–4 flowers. Even if we consider the locomotory cost of visiting these additional flowers, it seems that the total cost associated with APR for *G. soricina* are trivial and would not jeopardize its energy budget. However, APR activation decreased food intake of other vertebrates (*Aubert, Kelley & Dantzer, 1997*; *Vallès et al., 2000*) and so there may be an additional negative impact on the foraging behavior of *G. soricina*.

## CONCLUSIONS

The first direct measurements of the energy cost of APR activation in wild vertebrates were made for birds and were higher than for laboratory rodents suggesting that this response is less expensive for mammals (*Marais, Maloney & Gray, 2011*). However, the

large increase in RMR recently measured for the fish-eating Myotis (*Otálora-Ardila et al., 2016*; *Otálora-Ardila et al., 2017*) and the results of our study do not support this idea. The increase in RMR for the Pallas's long-tongued bat was one third of the greatest increase measured in the fish-eating Myotis but significantly higher than that measured for laboratory rodents and similar to the highest increase reported for birds. Our finding suggests that, like other features of APR, its metabolic cost might vary among bats in relation to ecological factors. Alternatively, body mass varies vastly among bat and mass might influence the metabolic cost of APR activation. For example, both diet and body mass of several species of bats were significantly related to leukocyte number, and bacterial killing ability (BKA) decreases with increasing roost permanence (*Schneeberger, Czirják & Voigt, 2013b*). In addition, pace of life might also affect immune response as BKA is negatively correlated with mass-adjusted basal metabolic rate (BMR) in birds (*Tieleman, Williams & Klasing, 2005*). BMR in bats varies with diet (*McNab, 2003*; *Cruz-Neto & Jones, 2006*) but its relation with immune response has not been examined. Measurement of the energetic cost of immune response in wild mammals is in its infancy and further work is warranted to evaluate its significance for the animal's energy budget.

## ACKNOWLEDGEMENTS

Christine Cooper, Noah Ashley and two anonymous reviewers kindly improved the quality of our manuscript.

### Funding

This work was supported by a grant from the bilateral program CNPq/CONACYT (# 490586/2013-9) to L. Gerardo Herrera M. and Ariovaldo P. Cruz-Neto. Lucía V. Cabrera-Martínez received a master grant from the Conselho Nacional de Pesquisa e Desenvolvimento Tecnológico (#161027/2014-8). L. Gerardo Herrera M. was supported by research grants from Dirección General de Asuntos del Personal Académico (DGAPA # IN202113) and by the PASPA-DGAPA program (#062-2014) of the Universidad Nacional Autónoma de México. Ariovaldo P. Cruz-Neto was funded by a grant from Fundação de Amparo a Pesquisa do Estado de São Paulo (# 2008/57687-0). The funders had no role in study design, data collection and analysis, decision to publish, or preparation of the manuscript.

### Grant Disclosures

The following grant information was disclosed by the authors:
Bilateral program CNPq/CONACYT: # 490586/2013-9.
Consatlho Nacional de Pesquisa e Desenvolvimento Tecnológico: #161027/2014-8.
Dirección General de Asuntos del Personal Académico: DGAPA # IN202113.
Universidad Nacional Autónoma de México: #062-2014.
Fundação de Amparo a Pesquisa do Estado de São Paulo: # 2008/57687-0.

## Competing Interests

The authors declare there are no competing interests.

## Author Contributions

- Lucia V. Cabrera-Martínez performed the experiments, authored or reviewed drafts of the paper, approved the final draft.
- L. Gerardo Herrera M. conceived and designed the experiments, performed the experiments, contributed reagents/materials/analysis tools, authored or reviewed drafts of the paper, approved the final draft.
- Ariovaldo P. Cruz-Neto conceived and designed the experiments, performed the experiments, analyzed the data, contributed reagents/materials/analysis tools, prepared figures and/or tables, authored or reviewed drafts of the paper, approved the final draft.

## Animal Ethics

The following information was supplied relating to ethical approvals (i.e., approving body and any reference numbers):

All protocols were performed under scientific collector license FAUT-0069 granted to L Gerardo Herrera M. by the Secretaría de Medio Ambiente y Recursos Naturales, Mexico.

## Data Availability

The raw data are provided as a Data S1.

## Supplemental Information

Supplemental information for this article can be found online at http://dx.doi.org/10.7717/peerj.4627#supplemental-information.

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
