# Peer review of "The energetic cost of mounting an immune response for Pallas’s long-tongued bat (Glossophaga soricina)"

_PeerJ, doi:10.7717/peerj.4627_

## Round 0.1 · original submission · Major Revisions

Please pay close attention to all the comments from the reviewers, and send in a point-by-point description of how you have addressed each comment when you send in your revised manuscript (should you decide to revise and resubmit)

·

Basic reporting

No comment

Experimental design

No comment

Validity of the findings

In abstract, it is difficult to compare APRs with other species without controlling for dose of LPS. This should be made clear in the abstract.

Additional comments

This study examines the effect of immune challenge (using bacterial lipopolysaccharide) upon energy expenditure and resting metabolic rates of the Pallas's long-tongued bat. The study is fairly straightforward and well-designed. The authors acknowledge several drawbacks of the study in the discussion: (1) Interspecific comparisons of APRs is somewhat misleading when different mass-corrected doses of LPS are used, and (2) body temperature was not measured. It would have been useful to know whether bats become hypo-or-hyperthermic from the LPS treatment vs. saline-treated bats. In small birds, LPS treatment can induce short-term hypothermia in some cases.

In regards to body mass changes from LPS, the authors compare their findings to other bat and birds. The authors only report house sparrows (Passer domesticus) in the discussion, but a number of other species have been examined (see my review paper on LPS-responses in birds. Ashley, N. T. and J. C. Wingfield. 2012. Sickness Behavior in Vertebrates: Allostasis, Life History Modulation, and Hormonal Regulation. In: G. E. Demas and R. J. Nelson, eds. EcoImmunology. Pp. 45-91. Oxford University Press).
. Also, there is pretty strong evidence that body mass loss is tied to initial body condition. Is there a method to assess body condition in bats?, and if so, this could help explain some of the variation in metabolic and body mass responses.

- The mean ambient temp was below the lower crit level of the animal… meaning all animals were thermogenic prior to injection and was wondering how much this would affect the estimates of RMR. Just something to think about...

Lines 276-LIne 280. This is a very long and confusing sentence. Please rephrase and break up.

LInes 286-288. What the same mass-corrected LPS dose in your study used in the fish-eating Myotis study? If not, differences in daily energy expenditure could be due to dose....

Line 293. ", then the results reveal"

Reviewer 2 ·

Basic reporting

Major Comments:
(1) The writing throughout the manuscript needs refining – as is, the writing presents some issues with readability due to basic, repeated grammatical errors. For example, the first sentence of the abstract:

“Activation of immune response has been long assumed to be an energy-costly process but direct measures of changes in metabolic rate after eliciting immune response disputes the universality of this assertion.”

This sentence should, as a first pass, be revised as follows:

“Activation of the immune response has long been assumed to be an energetically-costly process, but direct measures of changes in metabolic rate after eliciting an immune response disputes the universality of this assertion.”

However, further work to streamline the writing would really improve the manuscript:

“Mounting an immune response is thought to be energetically costly, however direct measures of metabolic rate during immune challenges contradict this assumption.”

These issues should be addressed throughout by careful editing and attention to grammar/sentence structure. I also found the logical order of sentences throughout difficult to follow. Again, in the abstract, the authors introduce the immune response and the assumptions about immune response activation in the first sentence (Lines 2-4). In the second sentence, authors define the immune response they’re addressing (the APR) and again state that it is energetically costly (lines 4-5). The connection between energetic cost and metabolic rate from the first sentence is not made explicit (only implicit), and the second sentence essentially repeats the same information. Authors can greatly improve the quality of the manuscript by carefully editing and revising the writing to address logical consistency and redundancies throughout.


Minor Comments:
1. It is not clear from the methods when body mass measurements were taken relative to treatment. Please clarify. Also, in discussing the changes in mass (line 222-223). The published report on the species of particular interest (M. vivesi) to the authors has issues in how their calculations were done overall – they suggest that bats lost weight in response to LPS injection, however their “pre-injection” mass measurements suggest that the change they are seeing is primarily explained by time of day and not the LPS injection (see Fig. 1 in Otalora et al 2016). So I don’t see this as a legitimate comparison. The Stockmaier paper also looked at 24 h rather than 10 h after injection. These differences in timing etc. should be made explicit because the LPS APR and recovery is so rapid (<48 h)

2. Throughout the results, authors are inconsistent on defining abbreviations when introduced in new sections (e.g., PI; line 202).

3. It’s unclear from the methods when animals were sampled. Were these data collected across a 30-day period, or were all individuals sampled on the same day? How did animals cope with captivity (body mass throughout this period)?

4. I find the Y axis label in Figure 1B confusing – “% Variation in MR after injection”. At first, I thought this meant, “how variable was the MR within an individual” – similar to heart rate variability – however it was later made clear to me that the authors meant “% increase in MR over baseline following injection”. This should be clarified.

5. In Table 1, I don’t understand the inclusion of the Mean body mass (Mean Mb) column. What does this metric tell us about? Mbi and Mbf seems sufficient.

Experimental design

No comments

Validity of the findings

The authors frame their work as an important comparative step, and contrast their data with data from small mammals, birds, etc. in an attempt to make broad generalizations about how different taxa respond to immune challenges. First, the authors need to be careful throughout to avoid over-generalizing. For example, the abstract, intro, and discussion state that:

“…large increase[sic] in metabolic rate after APR activation measured in piscivorous bat species … suggests that immune response is unusually costly for bats.” (Line 7-9)

I don’t think one species should be thought to represent the response of an entire (extremely diverse!) taxa, even if it’s the only published data out there, and the authors should not attempt to generalize to this degree.

More importantly though, there are some major problems with contrasting the manuscript under review with published literature (and therefore the presented premise of this project). First, different doses (and lots) of LPS are used across all these studies. The immune response to an LPS dose varies among lots, across different doses, parasite lode and exposure?, and also with life history stage/social context, etc. Authors acknowledge this fact briefly in their discussion (Line 264), however they do not address the fact that this variability largely undermines or at least limits comparative use of their study. I think this fact needs explicit attention throughout the manuscript. Due to the marked variation in response to LPS injection across these various contexts and among experiments, it is inappropriate and misleading to attempt to directly and quantitatively compare the data presented here with data from other studies. While qualitative comparisons that evaluate differences between the sexes, energetic states, or life history stages may be appropriate, the authors do not present analyses or discussion that could be used in such a context. The authors also state that their data collection and analytical methods for measuring metabolic rate differ from those in other studies. These difference further underscore that premising the entire paper on such comparisons is invalid. Finally, I would add that because doses are usually chosen by the ability to elicit a response (rather than some other external item), it's difficult to contrast among systems and the justification easily becomes circular.

Again, the authors introduce the idea that the differences in metabolic rate/cost of the immune response may be explained by “ecological factors” (Line 309-317). This is a really interesting idea, but the authors throw it into their conclusions without addressing it anywhere else, and they seem to suggest that this explanation makes the most sense without presenting any data from their own system to support this. It makes sense to me to bring this up as a contributing factor, but it remains unsubstantiated as a conclusion.

When these comparisons are appropriately limited and/or removed, these data are really only anecdotal. It is extremely difficult to assess whether these data provide any information to contrast this species with other studies, and there is too little data and the samples sizes are too small to address basic questions about the life history of this species or the evolutionary/taxonomic variation in the immune response.

Additional comments

As stated elsewhere, the data are interesting but stand largely as anecdotal. The scope of the article, as framed by the abstract, introduction, and discussion, needs to be substantially altered. Comparison of this work to others should remain qualitative, limited, and marked as speculative. Limitations of comparisons should be clearly discussed.

Reviewer 3 ·

Basic reporting

The authors have done an adequate job framing their study with relation to the literature. It is professionally written in relatively straightforward language (with a few exceptions) and the figures are sufficient and well-constructed. There are small typos and grammatical mistakes throughout that need attention.

There are some major style shifts throughout the manuscript that could be smoothed out. For example, the Introduction is really well constructed and is pleasant to read. The Abstract and Discussion, however, are much more disjointed and need the same kind of attention paid in setting up the manuscript in the Introduction. In the revisions please consider paying more attention to the overall flow and style of the manuscript.

Experimental design

Overall the experimental design is good. But there are several points that should be addressed, including a re-analyses of the entire data set.

1) the lack of inclusion of Ta into the analysis is a fundamental flaw to the data set. There was a large range of Ta measured during the study (at least 4 degrees -- but the minimum is never specified), including temperatures outside of the bat's reported TNZ. I suggest re-analysis of the data that tests for effects of Ta on MR.

2) The definition of the 'net RMR' is convoluted and needs to be clarified.

3) Instead of using Mb as a covariate, why not analyze mass-specific metabolic rates?

4) It is not clear how or in which software the data were analyzed.

Validity of the findings

The data appear to be robust, but see my above comments on analysis.

Hanging this around the 'large increase in RMR in fish-eating Myotis' seems a bit disingenuous, since in that study it is not clear if the bats were normothermic or thermoconforming at high temperatures. I think it it more interesting to rephrase this around the diversity of immunological adaptations that are emerging in bats and the strong selection on a reduced immunological tool kit.

The paragraph on L 245-280 in the Discussion meanders and could use substantial editing down to find the really important points.

Additional comments

I have made comments in the attached pdf.

Annotated reviews are not available for download in order to protect the identity of reviewers who chose to remain anonymous.

---

## Round 0.2 · Minor Revisions

Please note that there are still editorial flaws that could be strengthened if a colleague (who is a native English speaker) could read over the manuscript.

·

Basic reporting

The manuscript has improved editorially, but there are still minor editorial issues that need to be addressed. Perhaps having a colleague read who can assist with English language corrections would benefit the MS.

Experimental design

No comment

Validity of the findings

No comment

Additional comments

Overall, the authors have done a good job in responding to previous criticisms and toning down conclusions based upon previous studies.

Reviewer 2 ·

Basic reporting

The authors made effective changes to the writing in the manuscript.

Experimental design

The experimental design results in some important caveats to the interpretation, but these are now made clear by the authors in the methods and discussion.

Validity of the findings

The authors have made changes to the introduction and discussion that improves the validity of their conclusions.

Additional comments

Instead of placing the Ta analyses in the methods, where it can be easily missed, the authors shoudl move the analyses to the results section. They should also add information on the analyses run to their "statistical analyses" section.

There are a few typos throughout the corrections the authors made:

Line 52: “whithin” should be “within”

Line 64: add a space between “;” and “increase”

Reviewer 3 ·

Basic reporting

There are still a lot of grammatical error and awkward phrasing throughout. For example, the first 2 sentences of the abstract could be condensed to:

Mounting an immune response is thought to be energetically costly, however, direct measures of metabolic rate during immune challenges contradict this assumption.

Also, in the first paragraph of the introduction sentences 2, 3, 4, 5, 9 & 10 all start with a prepositional or introductory phrase (In particular, For example....). This is poor writing and should be revised.

Experimental design

The authors have met my original suggestions. Thank you.

166. Because you included random effects, you used generalised linear mixed effects models. Not just a GLM. Terminology matters here.
.

Validity of the findings

Looks good.

Additional comments

Thank you for taking the time to address my comments & those of the other reviewers. This will make a nice contribution to understanding the immune systems of bats and their overall variability.

Both Reviewer 2 & I raised issues with the author's discussion of injection dosages on 249-266 as one of the reason for not being 'strictly comparable with most of these studies for at least two reasons.' Especially with the laundry list after it, it minimizes that fact that species do vary incredibly in their response to LPS. It's just what happens and it's not just about body size as you present in your response to the reviewers. Without dosage trials or TK studies to find comparable physiological responses regardless of dosage level, this reasoning is not fair. I suggest minimizing this bit of discussion altogether or highlight _why_ this is true if you feel that a general audience won't intuitively understand it.

308-309. I believe that Brian McNab is the authority on this & shouldn't be ignored.

---

## Round 0.3 · accepted · Accept

Thanks for making thorough revisions.